# Tailored Synthesis of Catalytically Active Cerium Oxide for N, N-Dimethylformamide Oxidation

**DOI:** 10.3390/ma16020865

**Published:** 2023-01-16

**Authors:** Cedric Karel Fonzeu Monguen, En-Jie Ding, Samuel Daniel, Jing-Yang Jia, Xiao-Hong Gui, Zhen-Yu Tian

**Affiliations:** 1Institute of Engineering Thermophysics, Chinese Academy of Sciences, Beijing 100190, China; 2University of Chinese Academy of Sciences, Beijing 100049, China; 3School of Emergency Management and Safety Engineering, China University of Mining and Technology, Beijing 100083, China

**Keywords:** sol–gel method, cerium oxide, activation energy, DMF oxidation

## Abstract

Cerium oxide nanopowder (CeO_x_) was prepared using the sol–gel method for the catalytic oxidation of N, N-dimethylformamide (DMF). The phase, specific surface area, morphology, ionic states, and redox properties of the obtained nanocatalyst were systematically characterized using XRD, BET, TEM, EDS, XPS, H_2_-TPR, and O_2_-TPO techniques. The results showed that the catalyst had a good crystal structure and spherelike morphology with the aggregation of uniform small grain size. The catalyst showed the presence of more adsorbed oxygen on the catalyst surface. XPS and H_2_-TPR have confirmed the reduction of Ce^4+^ species to Ce^3+^ species. O_2_-TPR proved the reoxidability of CeO_x_, playing a key role during DMF oxidation. The catalyst had a reaction rate of 1.44 mol g^−1^_cat_ s^−1^ and apparent activation energy of 33.30 ± 3 kJ mol^−1^. The catalytic performance showed ~82 ± 2% DMF oxidation at 400 °C. This work’s overall results demonstrated that reducing Ce^4+^ to Ce^3+^ and increasing the amount of adsorbed oxygen provided more suitable active sites for DMF oxidation. Additionally, the catalyst was thermally stable (~86%) after 100 h time-on-stream DMF conversion, which could be a potential catalyst for industrial applications.

## 1. Introduction

N, N-Dimethylformamide (DMF) is a colorless transparent liquid which has been widely used as a chemical raw material [1]. DMF is miscible with water and most organic solvents and is known as a solvent in the battery industry (12%), polyacrylic fibre industry (22%), pharmacy and cosmetic industry (39%), polyurethane lacquers (17%), and pesticide formulations (10%) [2,3,4,5]. It is stable and difficult to biodegrade and has certain biological toxicity [6,7,8,9]. It is present in industrial wastewater as a nitrogen-containing volatile organic compound (NVOC). DMF typically has pungent odors that can harm the human body when inhaled into the respiratory system or through direct contact with the skin, as are its derivatives, such as NOx, which is a pollutant to the environment [10,11]. For this reason and for economic reasons, the transformation of DMF into harmless compounds has garnered increasing attention in recent years.

The general conversion methods for the NVOCs treatment are as follows: biotreatment (biotechnological purification) [1], activated carbon adsorption method [12], pervaporation separation [13], microbial degradation [14], electrochemical oxidation [15], and low-pressure pyrolysis [16]. However, the above methods require expensive conversion costs or complex post-treatment processes, which greatly limit the application of these methods of DMF conversion. Researchers have recently developed a method of catalytic oxidation of DMF using metal oxides [11,17,18]. Compared to other methods, metal oxide catalytic oxidation of DMF has the advantages of relatively low cost, easy operation, and high conversion efficiency. It involves reaction at lower temperatures and thus could be highly effective in oxidizing NVOC pollutants into harmless products, such as CO_2_, H_2_O, and N_2_ [11,19]. Therefore, this method is very suitable for the conversion of DMF. In recent years, some researchers have studied the catalytic degradation of NVOCs. Noble metals, such as Rh-, Ru-, Pd-, and Pt-based catalysts, have often been utilized for DMF elimination due to their superior low-temperature activity [10]. Among them, Ru has been reported to show the best performance [20,21]. However, because of the significant leaching of the active material induced by the coordination with the lone-pair electrons in the intermediate products, such as amine, that occurs during the oxidation reaction, these catalysts are not suggested for the combustion of DMF [1,22]. Nevertheless, noble metals are limited resources, and high costs have severely restricted their industrial applications. As transition metal oxides, cerium oxides appear to be good potential metal oxides because of their high oxygen storage capacity and rate of oxygen transfer, which are advantageous for their shape, giving CeO_x_ a positive effect during the oxidation reaction [23,24,25].

The main purpose of this study is to synthesize CeO_x_ nanocatalyst for DMF oxidation using the sol–gel technique. Particular attention has been paid to the structure, redox properties, and catalytic activity of the CeO_x_ nanopowder. XRD, BET, TEM, XPS, and UV–Vis were used to characterize the as-synthesized catalyst. Temperature-programmed and FTIR methods were utilized to investigate the thermal stability and redox properties. The catalytic performance of the powder for the DMF oxidation was evaluated in a fixed bed quartz reactor at room temperature. This aimed to correlate the activity, tailored structure, and physicochemical characteristics of CeO_x_.

## 2. Experimental

### 2.1. Catalyst Preparation

The catalyst is prepared using the sol–gel method, and the preparation process is shown in Appendix A and the literature [26]. Cerium nitrate hexahydrate (Ce(NO_3_)_3_.6H_2_O, 99.9%, Sinopharm Chemical Reagent Co, Ltd., Shanghai, China) was used as a precursor because it is easily soluble in water and ethanol. The preparation process was divided into six steps: (i) 6.0 g of cerium nitrate hexahydrate was dissolved in 60.0 mL distilled water at 60 °C and stirred in a beaker for 10 min; (ii) 1.0 g of sodium hydroxide was dissolved in 30.0 mL of distilled water at 60 °C and stirred for 10 min; (iii) the already prepared solutions were mixed and continuously stirred for 10 min. (iv) Furthermore, 100.0 mL of ethanol was added to the solution and stirred at 60 °C for 10 min to make it fully react. (v) Then, 15.0 mL of aqueous ammonia solution was added to the final mixture to adjust the pH value to 8. The mixture was then dried in an oven at 100 °C for 2 h to evaporate the solvent. (vi) Finally, the powdered sample was transferred to an electric furnace and calcined at 500 °C for 120 min at a rate of 5.6 °C min^−1^.

### 2.2. Catalyst Characterization

A variety of techniques were used to systematically characterize the prepared nanocatalyst. The phase information of the nanoparticles was analyzed using RIGAKU ULTIMA IV X-ray diffraction (XRD) at the scanning angle range of 5.0°−90.0°, and the scanning step was in the range of 0.5° using Cu Kα. The powder samples’ surface areas and textural characteristics were measured using an Autosorb IQ instrument (Quantachrome) at 77.4 °C to obtain a BET (Brunauer–Emmett–Teller). The morphology of the ceria oxides was observed via transmission electron microscopy (TEM) using a NIPPON Electronics 2100 microscope operated at 200 kV. The surface chemical composition of the sample was obtained via X-ray photoelectron spectroscopy (XPS) (Thermo Fly’s NEXSA) with an analytical voltage of 1486.8 eV and an analysis voltage of 15.0 kV. The redox properties were elaborated using temperature-programmed reduction (H_2_-TPR) and temperature-programmed oxidation ((O_2_-TPO) AutoChem Ⅱ 2920, Micromeritics). The optical properties were evaluated with a SHIMADZU UV−2600i visible–near-infrared spectrophotometer.

### 2.3. Catalytic Test

The catalytic performance of prepared Ce_2_O_3_ for DMF was analyzed in a fixed bed reactor under atmospheric pressure. The length of the reactor tube was 900 mm with an inner diameter of 8 mm. The device diagram of the catalytic test can be observed in Appendix A). An amount of 0.1 g of the CeO_x_ catalyst sample was placed into the reactor. The reaction gases were DMF and O_2,_ with Ar as the equilibrium gas. Ar was used as the carrier gas to move DMF into the reactor. The concentration of DMF was 1000 ppm, 10.0% O_2_ was diluted in Ar with a total flow rate of 50.0 mL min^−1^, and the corresponding gas space velocity (GHSV) was 30,000 mL g^−1^_cat_ h^−1^. A digital electric furnace controlled the temperature at increments of about 5 °C min^−1^. The exhaust gases were detected and analyzed using an FTIR spectrometer within a spectral range of 400–4000 cm^–1^. Before the catalytic test, the samples were pretreated in Ar (50.0 mL min^−1^) at 400 °C for 1 h to remove any surface-adsorbed substances. The conversion of the DMF is calculated using the following equation:DMF conversion (at temperature T) = [(X_in_ − X_out_)/X_in_] × 100(1)
where [X_in_] and [X_out_] represent the initial molar concentration of DMF at temperature T and the final molar concentration of DMF at the outlet, respectively.
(2)$${\mathrm{CO}}_{2}/\mathrm{NO}\;\mathrm{selectivity}\;(\%)=\frac{{\mathrm{CO}}_{2}/\mathrm{NO}\left(\mathrm{vol}.\%\right)}{\mathrm{Total}\;\mathrm{products}\;C\left(\mathrm{vol}.\%\right)}\times \;100$$

## 3. Characterization Results and Analysis

### 3.1. Catalyst Structure and Texture Analysis

Figure 1 shows the XRD patterns of the nanoparticles. The prepared sample has crystalline diffraction peaks. The as-prepared catalyst CeO_x_ corresponds to the cubic phase cerium oxide (CeO_2_) (JCPDS No.01-089-8436), which is in agreement with the literature [27,28]. The strong and sharp diffraction peak indicates the pure crystal structure of the sample.

The crystallite sizes (*D*) and micro-strain (*ε*) on the (111), (220), and (311) planes with the highest intensity were calculated using the Debye–Scherrer Formulas (3) and (4).
(3)$$D=\frac{0.9\lambda }{\beta cos\theta }$$
(4)$$\varepsilon =\frac{\beta }{2cos\theta }$$
where *β* is the diffraction broadening on the peak at half height for Bragg’s angle *θ* and *λ* represents the wavelength of the X-ray radiation. For more details on the calculation, see Table 1. The average grain size is 20.07 ± 0.2 nm, and the average microstrain is 0.12 ± 0.02%. With the increase in the crystallite size, the microstrain has a downward trend. This is because inherent defects may cause the microstrain in the crystal lattice, such as vacancies, spatial barriers, and concentration changes during the preparation process. The CeO_x_ nanocatalyst in this study is fine crystals, which are lower than 100 nm. The BET results show that the specific surface area of the sample is 3.8 ± 0.2 m^2^ g^−1^, with a pore volume of 0.01 ± 0.001 cm^3^ g^−1^ and a pore size of 25.7 ± 0.2 nm at the relative pressure of P/P_0_ = 0.98. Appendix A depict type 3 nitrogen adsorption–desorption isotherm and the pore size distribution, respectively. It is worthy to note that the pore size obtained in this work is quite smaller than those of the previously reported CeO_2_ (53.48 nm) [1] and CeO_2_-nanocube [23]. Small crystallite and pore size are advantageous for abundant oxygen species, playing a key role during DMF oxidation [20]. Therefore, the obtained CeO_x_ is expected to offer a good DMF conversion.

### 3.2. Morphology

To obtain the structural changes of the CeO_x_ catalyst due to calcination and reaction temperature, the surface morphology and bulk elemental constituent of the prepared nanopowder were characterized using TEM and EDS, as shown in Figure 2. The surface of the nanocatalyst has a spherelike morphology, which contrasts with the 3D ball-like shape of self-assembled M-CeO_2_ [11,23] and the large particles of CeO_2_ supported on multiwall carbon nanotube (CNTs) [1]. It can also be observed that the catalyst prepared has an aggregation of uniform small grain size. Looking at the EDS elemental distribution in Figure 2, it can be observed that the cerium and oxygen have similar distribution and local aggregation. The carbon arising from the carbon tape that was used as support during EDS analysis, as shown in Figure 2. It can be seen from the elemental distribution that cerium accounts for 53.60% of cerium, 24.20% carbon, and 22.20% oxygen. However, the 24% carbon is not considered because it arises from the carbon tape employed during EDS analysis. The observed spherelike morphology with small grain size agglomeration is in good line with XRD and BET results, which will positively affect the movement of absorbed oxygen and active sites advantageous for the oxidation reaction.

### 3.3. Elemental Composition and Ionic State

XPS was used to detect the surface chemical oxidation state of the Ce element. Figure 3 shows the deconvolution of the electron binding energy (BE) in the Ce 3*d* spectrum, while Table 2 shows the details of the distribution of Ce species. The XPS spectra of Ce 3*d* were divided into two spin–orbit components 3*d*_3/2_ and 3*d*_5/2_, using fityk 1.3.1 software (Copyright 2001-2015, Marcin Wojdyr). Ce 3*d*_5/2_ was deconvoluted into three peaks at 882.2, 883.9, and 888.6 eV, while Ce 3*d*_3/2_ was fitted into six peaks at 896.4, 898.1, 900.6, 900.8, 904.0, and 907.2 eV. The peaks at 883.9, 896.4, 900.6, and 904.0 eV are assigned to Ce^3+^, and the peaks at 882.2, 888.6, 898.1, 900.8, and 907.2 eV are assigned to Ce^4+^ [29,30]. Ce 3*d* mainly exists in the form of Ce^4+^ because the cerium in the catalyst is in the form of oxides. The ratio of Ce 3*d*_5/2_ is less than in Ce 3*d*_3/2_. The decrease in the binding energy causes the ratio of Ce^3+^/Ce^4+^ to decrease, which could be due to Ce^3+^ being converted to Ce^4+^.

Figure 4 is an XPS spectrum of the valence state and distribution of elements before the catalytic test, and Table 3 gives the detailed distribution of elements in the XPS spectrum. The O 1*s* peak was fitted into two to estimate the chemical states of different oxygen species. As shown in Figure 4, the O 1*s* core-shell exhibits two prominent peaks. The lower binding energy peak corresponds to lattice oxygen O_Lat_ (O^2−^) at 529.1 eV. The higher binding energy peak corresponds to the adsorbed oxygen (O_Ads_), expressed as OH^-^, and H_2_O at 532.5 and 534.7, respectively. These results are in agreement with the literature [30]. Surface oxygen is mainly composed of O_Lat_ and O_Ads_. O_Lat_ exists in the form of oxygen combined with transition metal elements, and O_Ads_ mainly exists in the form of hydroxide ions and oxygen ions. The proportion of O_Ads_ on the sample surface is higher than that of O_Lat_. The high proportion of adsorbed oxygen could enhance the catalyst’s ability to react with the surface products to form products such as H_2_O, NO_2_, and CO_2_, which improves the mass transfer driving force of product diffusion, thus improving the reaction rate.

### 3.4. Redox Properties

H_2_-TPR studies the reducibility of the prepared catalysts, and the results of the H_2_ consumption by CeO_x_ as a function of temperature are shown in Figure 5a and Table 4. With the increase in reduction temperature, three hydrogen consumption peaks with different strengths were observed: a small peak at 575.6 °C, the second one at 589.9 °C, and a third wide peak appear at 641.8 °C. The temperature peak at <600 °C is due to the reduction of Ce^4+^ to Ce^3+^ on the cerium oxide surface with oxygen vacancy, while the one > 600 °C represents a reduction of Ce^4+^ to Ce^3+^ by H_2_ inside the bulk oxygen from cerium oxides species [1,31]. Therefore, the H_2_-TPR profiles show that the CeO_x_ could result from the removal of bulk oxygen of cerium oxide.

The oxygen mobility in the CeO_x_ was studied using O_2_-TPO. Figure 5b illustrates the O_2_ consumption as a function of the temperature. The peak areas of the TPO profiles over the synthesized catalysts represent oxygen consumption at 693.4 °C. The calculated peak area is listed in Table 4. As shown in Figure 5b, the O_2_ consumption is high on CeO_x_. This indicates that oxygen is not consumed. The reoxidation initiates at ~300 °C, and the catalyst completely recovered at around ~693.4 °C. These findings are in good agreement with previous studies [32,33]. In summary, TPR and TPO results reveal that CeO_x_ could be more active from the range of 300 to 700 °C for DMF oxidation.

### 3.5. Optical Properties

The adsorption spectrum of the cerium oxide catalyst, as well as the Tauc’s plot displaying the estimated bandgap energy (*E_g_*), were investigated in the region of 200–800 nm, and the results are illustrated in Figure 6. Figure 6a shows an adsorption’s decrement as the wavelength extends to the visible region. The intense peak referring to the strong absorbance appeared at about 300–350 nm. According to the literature, this peak is related to the charge transfer from O 2p to Ce 4f, which overruns the well-known f-f spin–orbit splitting of the Ce 4f state [34,35]. The estimation of the optical bandgap energy *E_g_*, can be evaluated by the following equation:*αhν* = *A* (*hν* − *E_g_*)*^n^*(5)
where *A* is a refractive index constant and *n* is a constant related to the transition’s nature (in this work, *n* = 2 considering the cerium oxide presented a direct bandgap).

The corresponding Tauc plot of (αhν)^2^ is presented in Figure 6b. The *E_g_* for the catalyst was observed from the straight line’s intersection with the photon energy hν axis. *E_g_* was determined to be 2.93 ± 0.05 eV. This value is smaller compared to bandgap energies of cerium oxide nanoparticles prepared in the literature from other preparation methods, such as coprecipitation [36], microwave [35], and microwave-assisted hydrothermal [37]. As previously reported, small *E_g_* facilitates the movement of O_Lat_ at the catalyst’s surface, enhancing the catalytic activity [26]. Therefore, the catalyst in this work is expected to exhibit a good conversion during DMF oxidation.

### 3.6. Catalytic Test

The catalytic performance of CeO_x_ for DMF oxidation was studied in the temperature range of 100–400 °C. Figure 7a shows the catalytic behavior of DMF on the prepared CeO_x_ catalyst. As shown in Figure 7a, the catalyst had little effect on the conversion of DMF before 300 °C. The conversion of DMF gradually increased from 300 °C and reached 81.48 ± 2% at 400 °C. In addition, Figure 7b shows the selectivity of other products—CO_2_ and NO_2_—produced by the reaction at 400 °C. The main products are CO_2_ and NO_2_, with selectivities of 62.48 ± 2% and 37.52 ± 2%, respectively. As illustrated in Figure 8 and Appendix A, it is worth noting that the as-prepared catalyst retained its catalytic activity and reproducibility after three running tests, indicating that it is reusable for DMF oxidation. The DMF conversion in this study showed a higher conversion compared to those reported for Cu Ru/C (73.00%) [38], Pd/C (51.00%) [39], Ru/C (50.00%) [40], and Cu/ZnO/Al_2_O_3_ (40.60%) [41].

According to the XRD analysis, CeO_x_ has a small crystallite size, which is beneficial for oxidation reactions [42]. The performance of CeO_x_ is related to the spherical and smooth structure and the agglomeration of fine particles (see TEM results), which are conducive to enriching the movement of lattice oxygen. XPS results further confirmed the abundance of O_Lat_. This catalytic behavior can be attributed to the good content of O_Lat_ and the small amount of loading detected on the catalyst surface. In addition, the catalyst presented a large number of Ce^4+^ species, which was proven to have a positive effect on DMF oxidation [43]. In this work, CeO_x_ nanopowder was active between 300 and 400 °C, confirming the H_2_-TPR and O_2_-TPO results. Therefore, the catalytic activity of CeO_x_ can be attributed to the above properties.

Furthermore, the catalytic performance of the as-prepared catalysts was compared based on the specific reaction rate and the apparent activation energy, as illustrated in Figure 9, which were calculated using the Arrhenius equation in the DMF conversion range within 15% [44,45]. The following equations were used to calculate the reaction rate and the apparent activation energy:*k* = *A* exp (−*E_a_*/*RT*)(6)
*r* (mol g^−1^ s^−1^) = *F*_0_ *X*/W_cat_(7)
where *k* is the rate constant, *E_a_* is the apparent activation energy, *R* is the gas constant, *T* is the temperature, *A* the pre-exponential factor, *F*_0_ refers to the molar flow rate, *X* is the conversion achieved at a certain temperature, and W_cat_ is the weight of the catalyst. DMF’s oxidation started as the reaction rate increased with a gradual temperature increase, as shown in Figure 9a. The *E_a_* in this work was found to be 33.30 ± 3 kJ mol^−1^, which is smaller than those of CeO_2_-nanorods (37.48 kJ mol^−1^) [23], CeO_2_-nanocubes (47.32 kJ mol^−1^) [23], and CeO_2_/CNTs-M (72.97 kJ mol^−1^) [1] reported in the literature. Previous studies revealed that low *E_a_* is beneficial for better catalytic activity [23,46]. Therefore, the low *E_a_* obtained by CeO_x_ is expected to play a key role during the DMF conversion.

Furthermore, it is widely recognized that catalytic stability is an important parameter for industrial applications. The stability test on the DMF conversion was performed over 100 h at a constant temperature setting of 400 °C, as shown in Figure 10. The catalyst showed good durability during a continuous DMF conversion, with no significant deactivation after 100 h time-on-stream at 86%. However, there is a negligible performance decrease around 14% with continuous time-on-stream. The slight deactivation of the CeO_x_ during the stability process might be due to the following reasons: (1) active oxygen vacancy sites decrease due to catalyst sinter and/or (2) disintegration of the active CeO_2_ crystal phases, which might cause the catalyst deactivation. Alternatively, coke formation over solid catalysts from DMF could lead to continuing oxidation at high temperature.

The DMF oxidation reaction pathway over CeO_x_ is proposed based on the experimental data, as shown in Figure 11, in accordance with the previous literature [20,22,47]. The first step is the breakage of the C-N bond in DMF via hydrothermal decomposition, which leads to H-attraction, producing HCOOH and HN(CH_3_)_2_. Then, the HCOOH will be further oxidized to CO_2_ and H_2_O, and the HN(CH_3_)_2_ will also gradually be oxidized using the lattice oxygen to produce NO_2_, H_2_O, and CO_2_, which agree with the final products detected (i.e., NO_2_, H_2_O, and CO_2_).

## 4. Conclusions

A cerium-based nanocatalyst for DMF oxidation was synthesized in this work using the sol–gel method. Structure, morphology, elemental distribution and the optical properties of the synthesized CeO_x_ were characterized using XRD, TEM-EDS, XPS, and UV-visible spectroscopy. The crystal structure shows that CeO_x_ has a smaller grain size, and the aggregation of spherical and fine particles is observed via TEM analysis. XPS analysis showed that CeO_x_ samples exhibited a large amount of beneficial adsorbed oxygen. Due to the combined effect of small grain size, high O_Ads_ content, and highly dispersed Ce^4+^ on the catalyst surface, the DMF conversion reached ~82 ± 2%. CeO_x_ possessed a reaction rate of 1.44 mol·g^−1^_cat_·s^−1^ and apparent activation energy of 33.30 ± 3 kJ mol^−1^. CeO_x_ is stable after 100 h time-on-stream of DMF conversion. A mechanism was proposed to understand the decomposition of DMF to form products such as NO_2_ and CO_2_. The overall results of this work show that CeO_x_ provides a more suitable active site for DMF oxidation and could be a potential catalyst for further industrial applications.

## Figures and Tables

**Figure 1 materials-16-00865-f001:**
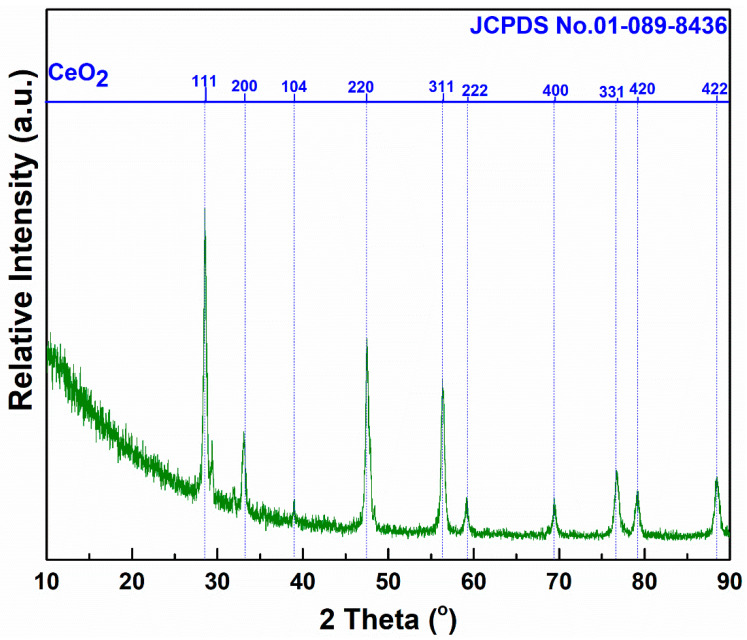
XRD pattern of the catalyst.

**Figure 2 materials-16-00865-f002:**
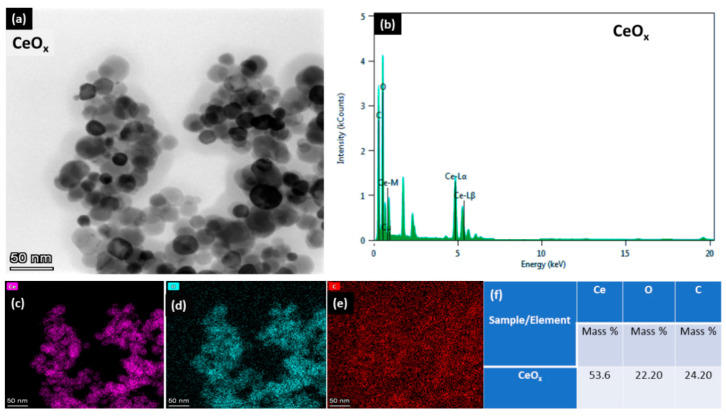
TEM image (**a**), elemental spectra (**b**), EDS mapping (**c**–**e**), and relative atomic of the elements in the as-synthesized catalyst (**f**).

**Figure 3 materials-16-00865-f003:**
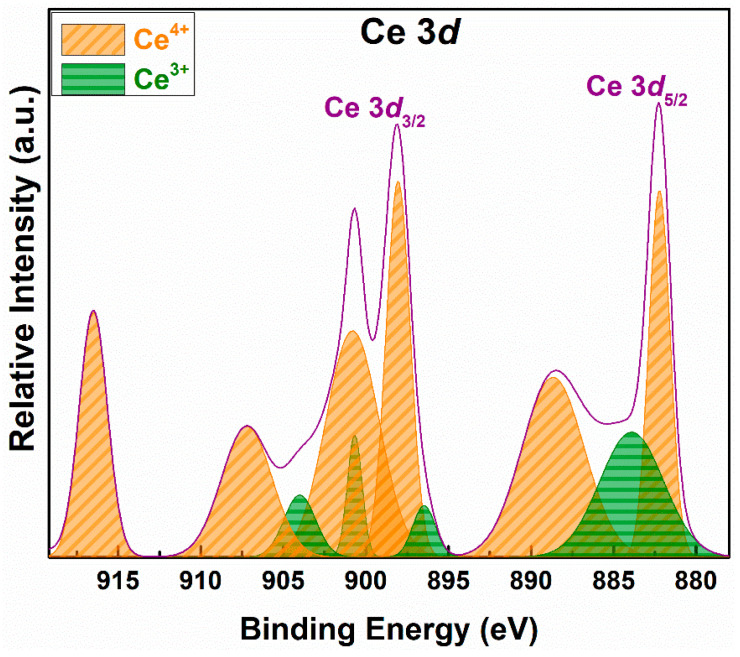
Deconvolution spectrum of Ce 3*d* of the catalyst.

**Figure 4 materials-16-00865-f004:**
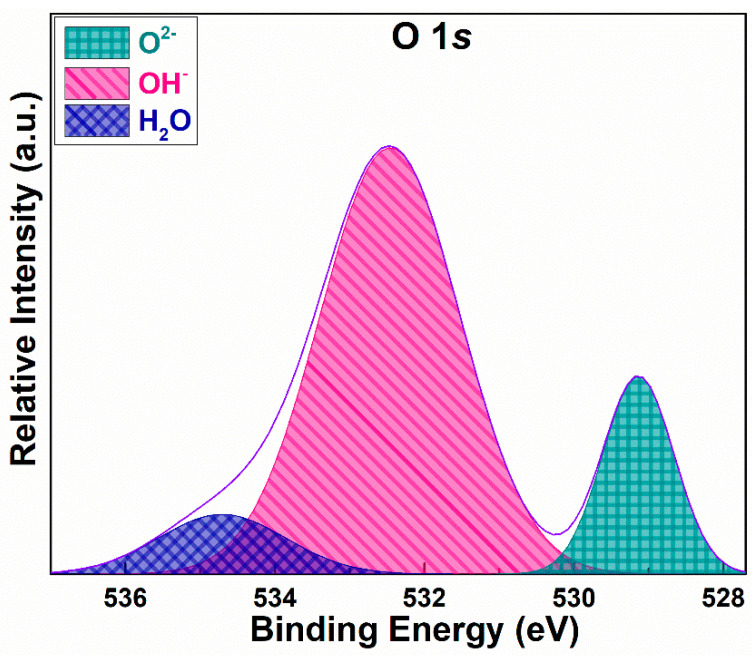
O 1*s* spectrum of the catalyst.

**Figure 5 materials-16-00865-f005:**
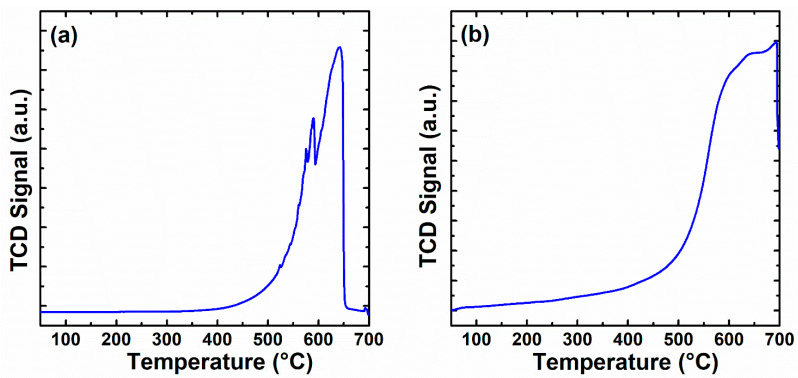
H_2_-TPR profile (**a**) and O_2_-TPO (**b**) of the catalyst.

**Figure 6 materials-16-00865-f006:**
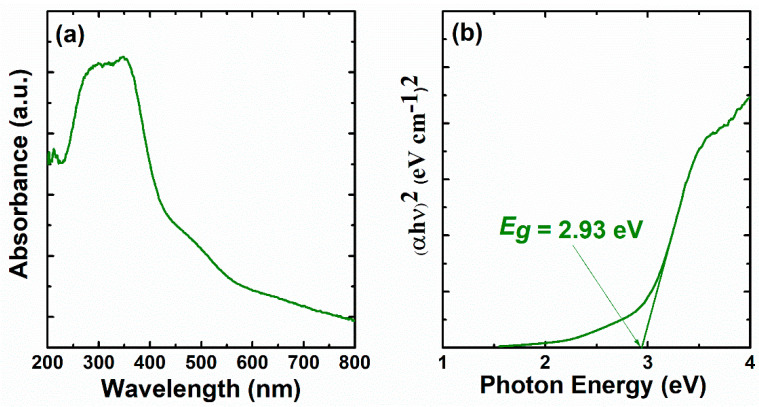
UV–Vis spectrum of sample (**a**), and Plot of (αhν)^2^ versus photon energy (**b**).

**Figure 7 materials-16-00865-f007:**
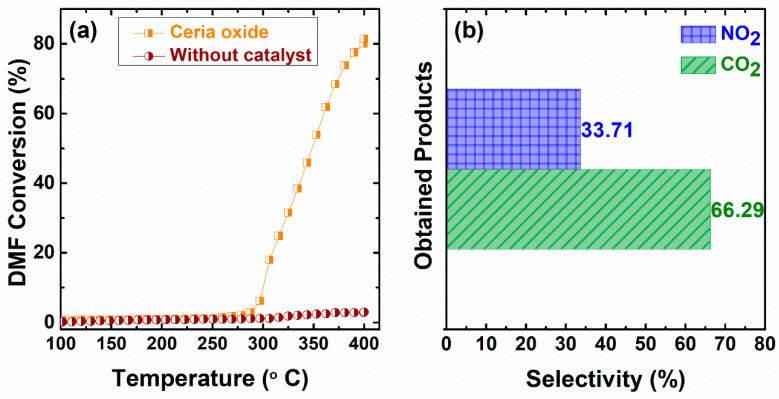
Conversion of DMF oxidation on the catalyst (**a**) and selectivity of the obtained product (**b**).

**Figure 8 materials-16-00865-f008:**
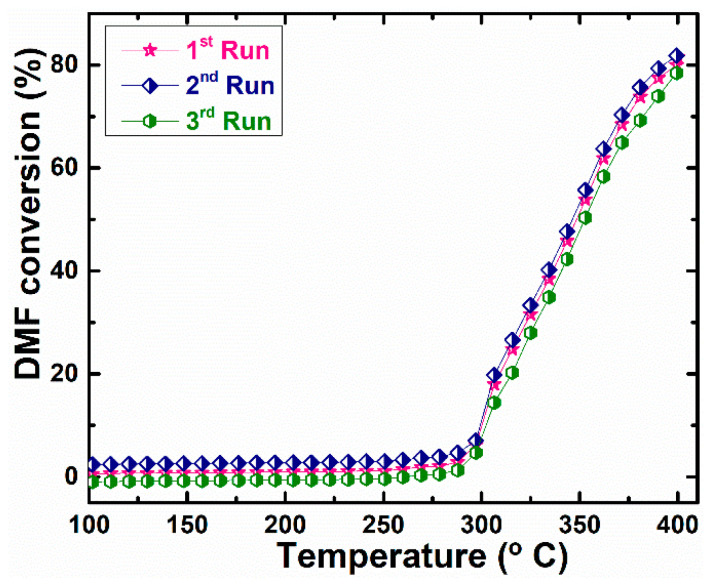
Reproducibility of DMF conversion over CeO_x_ catalyst.

**Figure 9 materials-16-00865-f009:**
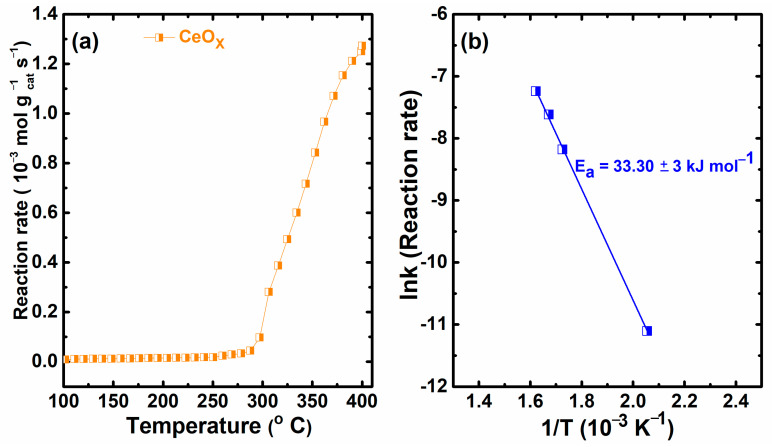
Variation of the reaction rate of DMF (**a**) and Arrhenius plot for the catalyst on the DMF reaction evaluated from data in the conversion range within 15% (**b**).

**Figure 10 materials-16-00865-f010:**
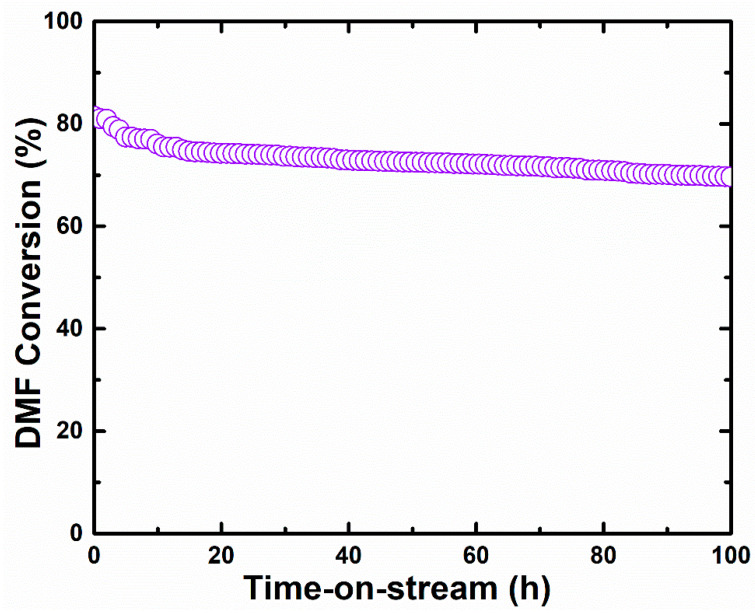
Time-on-stream conversion of DMF catalyzed by cerium oxide after 100 h.

**Figure 11 materials-16-00865-f011:**
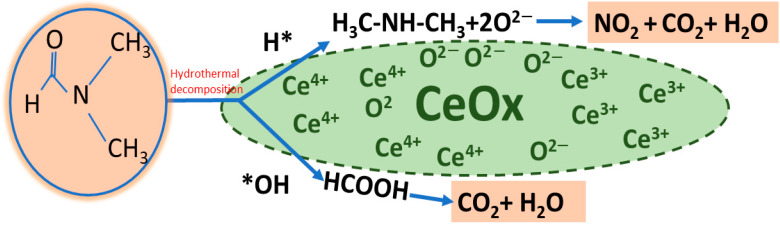
Proposed oxidation pathways of DMF oxidation over supported CeO_x_ catalysts.

**Table 1 materials-16-00865-t001:** Grain size and microstrain of the catalyst.

Catalyst	hkl	FWHM (*β*)	2*θ* (º)	Crystallite Size *D* (nm)	*D* Average (nm)	Microstrain ε (%)	ε Average (%)
CeO_x_	111	0.42	28.55	19.52	20.07	0.14	0.12
	220	0.43	47.55	20.19		0.13	
	311	0.44	56.38	20.49		0.11	

Note: hkl refers to Miller indicies, *θ* refers to Bragg’s angle, and FWHM refers to the full width at half maximum of the peak.

**Table 2 materials-16-00865-t002:** Deconvoluted XPS spectrum of Ce 3*d* distribution in the catalyst.

Catalyst	Parameters			Ce 3*d*5/2				Ce 3*d*3/2			
CeO_x_	Species	Ce^4+^	Ce^3+^	Ce^4+^	Ce^3+^/Ce^4+^	Ce^3+^	Ce^4+^	Ce^3+^	Ce^4+^	Ce^3+^	Ce^4+^
	BE (eV)	882.2	883.9	888.6	0.42	896.4	898.1	900.6	900.8	904.0	907.2
	RA (%)	28.97	29.58	41.45		3.58	27.84	5.51	36.78	6.04	20.25

Note: BE refers to binding energy, and RA refers to the relative area of the peak.

**Table 3 materials-16-00865-t003:** Deconvoluted XPS spectrum of O 1*s* distribution in the catalyst.

Catalysts	Parameter	O 1*s*			
CeO_x_	Species	O^2−^	OH^−^	H_2_O	O_Lat_/O_Ads_
	BE (eV)	529.1	532.5	534.7	0.21
	RA (%)	17.31	73.38	9.31	

**Table 4 materials-16-00865-t004:** H_2_-TPR profiles and O_2_-TPO peaks data.

Cat.	H_2_ Consumptions (mmol/g)	O_2_ Consumptions (mmol/g)
Peak 1	Peak 2	Peak 3	Total	Peak 1	Total
CeO_x_	4.03	1.31	6.52	11.86	7.08	7.08
575.6 °C	589.9 °C	641.8 °C		693.4 °C

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
