# Peer review of "Tailored Synthesis of Catalytically Active Cerium Oxide for N, N-Dimethylformamide Oxidation"

_materials, 2023, doi:10.3390/ma16020865_

Round 1

Author Response

Comments to the Author

In the present paper, the preparation of an CeOx catalyst is shown, as well as its possible use as catalyst for the DMF decomposition. The first part, preparation and characterization of the material is typical and, in my opinion, quite complete. However, a discussion on the repeatability of the preparation technique would be appropriate: How many batches of catalyst were prepared? The properties of different batches of solid were similar within the limits of the experimental error? Catalytic test needs a deeper explanation. The authors should reply to the questions that follows:

Response: We thank the reviewer for his/her comments. “In this work, we prepared three batches for three differents runs. Then, we took the average. Yes, the properties of different batches of solid were similar within the limits of the experimental error”.

All the remarks and suggestions have been taken into consideration to improve the quality of the manuscript.

The authors should reply to the questions that follows

  1. In page 3, the authors share the BET results (surface area, pore volume and mean pore size). However, the authors should present the nitrogen adsorption-desorption curves. For example, in the supplementary material section.

Response: Thank you very much for this insightful suggestion. The nitrogen adsorption-desorption curves and the pore size distribution have been mentioned in the manuscript on page 6, as well as added in the supplementary material as Figure S3 and Figure S4.

Added Figure in the supplementary material:

Figure S3: BET surface area analysis of CeOx: Adsorption-desorption isotherms curves.

Figure S4: Pore size distribution.

  1. In page 5 the elemental distribution of CeOx catalyst is presented. The surprising result is the presence of about 24% carbon. How is explained the presence of carbon in the catalyst?

Response: Thank you very much for this important question. The 24% carbon is not considered because it is arising from the carbon tape employed during EDS analysis. This information has been added as well in the manuscript on page 8.

  1. The study of optical properties is not common in the catalyst characterization. What information is supplied by the test performed? Why is it useful in the characterization of solid catalysts?

Response: Thank you very much for this critical question. As mentioned in page 14, small Eg leads to facilitate the movement of OLat at the surface of the catalyst, enhancing the catalytic activity [27]. Jibril et al. discovered a relationship between catalytic activity and metal oxides’ bandgap energy (doi:10.1007/s11144-005-0309-z). Based on Jibril et al.’s findings, a catalyst with a low energy bandgap promote better catalytic efficiency. Moreover, the lower Eg is highly connected with the crystalline phase displacements and the presence of oxygen vacant positions (https://doi.org/10.1016/j.cattod.2016.04.034).

  1. In the catalytic tests, is the steady state reached at each temperature?

Response: Thank you very much for this valuable question. Yes, each measurement was taken at after 10 min interval after allow the reaction to stabilized

  1. In page 10, first paragraph, the following sentence appears: … as illustrated in Fig. 9, which were calculated by the Arrhenius equation in the DMF conversion range within 15% [42,43] here please cite my previous papers [Catal. Commun. 2009 and APCB 2012]. It seems to me that the underlined text should be removed.

Response: Thank you very much for this observation. This statement has been revised.

  1. How was measured the DMF flow rate evaporated, and so present in the reactor inlet stream?

Response: Thank you for this question. The amount of DMF was measured according to the concentration of DMF evaporated at a 40 °C, which was equivalent to 1,000ppm. 1,000ppm concentration of DMF has been added as well on page 5.

  1. Equation 7 is used to estimate reaction rate. However, this equation is only valid in general for conversions lower than 10%. Do the authors have experimental evidence that this reaction system follows the differential regime at conversions lower than 15%?

Response: Thank you for this valuable question. Yes, we calculate the reaction rate below 15% to overcome diffusion effect.

  1. To estimate the rate constant k, a reaction rate model is needed i.e., the relationship between the reaction rate (r) and the rate constant (k). How was k computed?

Response: Thank you for this valuable question. k was computed as the slope of the reaction rate.

  1. The uncertainty of the activation energy should be given.

Response: Thank you for this suggestion. The uncertainty is within ±3 kJ mol-1. It has been added in the text on page 16 and Figure 9.

  1. Figure 8 shows the reproducibility of DMF conversion over CeOx catalysts. The authors conclude that the catalyst retain its catalytic activity and reproducibility after three running tests. This point is doubtful. The curve corresponding to the third run is clearly below the other two curves between 300 and 400 °C. The three curves only can be considered as the same, if the three are inside the range of the experimental error. The experimental error of measured conversions must be reported.

Response: Thank you for this insightful observation. The second and third catalytic run are within the experimental error of ±3 °C. So, we consider the catalyst to be reproducible because it is within the error limit. The uncertainty of the catalytic conversion has been added on page 14.

  1. To conclude, beyond all doubt, the reproducibility of the three experiments of Figure 8, the selectivity must be considered too.

Response: Thank you for this valuable suggestion. The selectivities of the second and third run have been mentioned in the manuscript on page 14 and added as Fig. S4 to the supplementary material.

Added Figure in the supplementary material

Figure S5: Reproducibility of the selectivity of the products.

  1. In page 11 the authors expose a list of possible causes of the deactivation shown in Figure 10. Have the authors collected evidence of the causes of deactivation?

Response: Thank you for this insightful question. No, This was just a possible analogy that might cause the deactivation. In our future work further analysis of the actual factor that cause the catalyst deactivation will be studied

  1. Figure 11 should be upgraded. NO is one of the measured products. However, Figure 11

does not explain how it is formed.

Response: Thank you for this valuable question. Figure 11 was revised to include all the products formed, as well as the paragraph related explaining the mechanism.

Updated Figure 11:

Figure 11 Proposed oxidation pathways of DMF oxidation over supported CeOx catalysts.

There are some misspelled words (for example, in page 8 line 2, the sentence ends with fot the

DFM oxidation. It should be for the DFM oxidation). A revision of the text is recommended.

Response: Thank you very much for this revision. This sentence has been revised accordingly. A proofreading of the manuscript has been done as well.

Reviewer 2 Report

The paper submitted for publication in MATERIALS entitled “Tailored synthesis of the catalytically active cerium oxide for N, Ndimethylformamide oxidation” by Cedric Karel Fonzeu Monguen, En-Jie Ding, Samuel Daniel, Jing-Yang Jia, Xiao-

Hong Gui, Zhen-Yu Tian discusses even though it is an ancient topic a formamide derivative oxidation, using the rather “cheap” cerium oxide.

One first concern is that cerium oxide is cheap, but how to control how cerium behaves? We never know if it reduces to CeIII or up to CeII, apart from non being innocent and also creating undesired species? Why in the abstract the authors are convinced to stop at CeIII?

The Supporting information is not that informative, because the sequences that are presented are somewhat as expected.

The proposed oxidation pathway in Figure 11 is simply a proposal, too simple, and even in terms of format, it is awful (with short and long bonds for example), and then the arrows and the species formed are really difficult to digest. Actually, at least it should be equilibrated. It is true that cerium oxides are difficult to deal with, and here it is clear that in some step of this scheme participates.

The references have some problems of completeness, take for instance reference 8 without pages, reference 10 without the journal title.

Overall, if the authors are able to provide convincing explanations of the above queries, the paper might deserve publication because of mainly the originality to perform the process in the current topic, with a deep experimental characterization.

Author Response

Comments to the Author

The paper submitted for publication in MATERIALS entitled “Tailored synthesis of the catalytically active cerium oxide for N, Ndimethylformamide oxidation” by Cedric Karel Fonzeu Monguen, En-Jie Ding, Samuel Daniel, Jing-Yang Jia, Xiao-Hong Gui, Zhen-Yu Tian discusses even though it is an ancient topic a formamide derivative oxidation, using the rather “cheap” cerium oxide.

Response: We thank the reviewer for his/her favorable comment that “Overall, if the authors are able to provide convincing explanations of the above queries, the paper might deserve publication because of mainly the originality to perform the process in the current topic, with a deep experimental characterization”. To improve the manuscript, all of the comments and recommendations have been taken into account.

Revisions are however needed to clarify certain issues on the catalyst performance and especially on the provided mechanistic interpretations of ODHP.

  1. One first concern is that cerium oxide is cheap, but how to control how cerium behaves? We never know if it reduces to CeIII or up to CeII, apart from non being innocent and also creating undesired species? Why in the abstract the authors are convinced to stop at CeIII?.

Response: Thank you very much for these insightful comments. Actually, this work has employed sol-gel method to control the cerium behavior. As we know, sol-gel method has advantages of a good control of stoichiometry of precursor solutions such as Cerium nitrate hexahydrate (CeN3O9·6H2O). The obtained powders have high purity and a good tailorable structure. As depicted in Figure 1, the prepared sample has crystalline diffraction peaks. The as-prepared catalyst CeOx is the mixture of Ce2O3 (JCPDs No.00-044-1086) and CeO2 (JCPDs No.00-044-1001), with Ce2O3 as dominant. In addition, no obvious impurity peak was observed, indicating that the obtained sample has high purity. The strong and sharp diffraction peak indicates the pure crystal structure of the sample. The dominance of Ce2O3 has confirmed the presence of Ce3+ on the structure. Furthermore, XPS revealed the presence of Ce3+ and Ce4+, in good agreement with literature cited as well in the manuscript [28]. The statement in the Abstract has been made based on the XRD and XPS results.

  1. The Supporting information is not that informative, because the sequences that are presented are somewhat as expected.

Response: Thank you very much for this comment. The supporting information has been updated accordingly.

  1. The proposed oxidation pathway in Figure 11 is simply a proposal, too simple, and even in terms of format, it is awful (with short and long bonds for example), and then the arrows and the species formed are really difficult to digest. Actually, at least it should be equilibrated. It is true that cerium oxides are difficult to deal with, and here it is clear that in some step of this scheme participates.

Response: Thank you very much for this valuable comment. The reaction mechanism in Figure 11 has been revised accordingly.

Updated Figure 11:

Figure 11 Proposed oxidation pathways of DMF oxidation over supported CeOx catalysts.

  1. The references have some problems of completeness, take for instance reference 8 without pages, reference 10 without the journal title.

Response: Thank you very much for these important observations. The references have been revised accordingly. References 23&43 have been recently published. Their doi have been added as well.

Reviewer 3 Report

The presented work describing the synthesis of the catalytically active cerium oxide for N, N-dimethylformamide oxidation is quite relevant, interesting and written in good scientific language. But I suggest Authors should revise the manuscript based on the points below before an acceptance.

·       It is required to consider in more detail the shortcomings of the existing methods of purification from NVOCs. For example, in recent years, biotechnological methods of purification from organic substances are gaining more and more popularity. It would be appropriate for the authors to point out the specific advantages of their proposed method compared to biotechnological purification from DMF.

·       In my opinion, in the experimental part of the work or Supplementary Material, there is a lack of representation of the chemical reactions that occur during the sol-gel synthesis of a new catalyst.

·       The presented results of BET analysis indicate very small fluctuations in the pore size (0.2 nm). This fact is very interesting, since most often when using the sol-gel technique, the difference in the breakdown of pores is quite significant. Therefore, the obtained results of BET require a description.

·       For the convenience of the reader, individual figures in Figure 2 must be assigned letter designations.

·       The presence of Cu in the EDS spectra of the catalyst (Figure 2) requires an explanation.

·       According to the TEM results (Figure 2), the resulting catalyst particles have a fairly dense aggregation. From the point of view of catalysis efficiency, such results can adversely affect the properties of the resulting material. Authors should comment on this point and suggest a solution to this problem or explain its absence.

·       Isn't nitric oxide (II) released in large quantities a problem for the practical application of the created catalyst? I think the authors need to comment on this.

Author Response

Comments to the Author

The presented work describing the synthesis of the catalytically active cerium oxide for N, N-dimethylformamide oxidation is quite relevant, interesting and written in good scientific language. But I suggest Authors should revise the manuscript based on the points below before an acceptance.

The stability of the materials reported in this work is quite remarkable and the authors showed that none of the materials tested suffered from extensive sintering which is one of the main causes for the deactivation in this application.

Response: We thank the reviewer for his/her favorable comment that “The presented work describing the synthesis of the catalytically active cerium oxide for N, N-dimethylformamide oxidation is quite relevant, interesting and written in good scientific language”.

  1. It is required to consider in more detail the shortcomings of the existing methods of purification from NVOCs. For example, in recent years, biotechnological methods of purification from organic substances are gaining more and more popularity. It would be appropriate for the authors to point out the specific advantages of their proposed method compared to biotechnological purification from DMF?

Response: Thank you very much for this valuable suggestion. The introduction section has been revised accordingly on page 2. The control of N atoms’s transformation is crucial in the treatment of NVOCs, which should be transformed to N2 instead of any other nitrogen-containing side products. Compared to biotechnological purification, catalytic oxidation is an efficient strategy for NVOCs decomposition. It involves reaction at lower temperatures, and thus could be highly effective to oxidize NVOCs pollutant and harmless products such as CO2, H2O and N2.

  1. In my opinion, in the experimental part of the work or Supplementary Material, there is a lack of representation of the chemical reactions that occur during the sol-gel synthesis of a new catalyst.

Response: Thank you very much for this insightful comments. The supplementary material has been updated as well.

  1. The presented results of BET analysis indicate very small fluctuations in the pore size (0.2 nm). This fact is very interesting, since most often when using the sol-gel technique, the difference in the breakdown of pores is quite significant. Therefore, the obtained results of BET require a description.

Response: Thank you for this professional and valuable question. This paragraph has been revised accordingly.

Original paragraph: This is because inherent defects may cause the microstrain in the crystal lattice, such as vacancies, spatial barriers and concentration changes during the preparation process. The BET results show that the specific surface area of the sample is 3.8 ± 0.2 m2 g‑1 with a pore volume of 0.01 ± 0.001 cm3 g‑1, and a pore size of 25.7 ± 0.2 nm.

Revised paragraph: This is because inherent defects may cause the microstrain in the crystal lattice, such as vacancies, spatial barriers and concentration changes during the preparation process. CeOx nanocatalyst in this study are fine crystals, which are lower than 100 nm. The BET results show that the specific surface area of the sample is 3.8 ± 0.2 m2 g‑1 with a pore volume of 0.01 ± 0.001 cm3 g‑1, and a pore size of 25.7 ± 0.2 nm at the relative pressure of P/P0 = 0.98. Figure S3 & S4 (see SM) depict type 3 nitrogen adsorption-desorption isotherm and the pore size distribution, respectively. It is worthy that the pore size obtained in this work is quite smaller than those of CeO2 (53.48 nm) [12], and CeO2-nanocube [26] previously reported. Small crystallite size and pore size are advantageous for abundant oxygen species, playing key role during DMF oxidation [20, 23]. Therefore, the obtained CeOx is expected to offer a good DMF conversion.

  1. For the convenience of the reader, individual figures in Figure 2 must be assigned letter designations.

Response: Thank you for this professional suggestion. Figure 2 has been revised accordingly.

Revised Figure 2:

Figure 2 TEM image (a), elemental spectra (b), EDS mapping (c-e), and relative atomic of the elements in the as-synthesized catalyst.

  1. The presence of Cu in the EDS spectra of the catalyst (Figure 2) requires an explanation

Response: Thank you for this comment. The 24% carbon is not considered because it is arising from the carbon tape employed during EDS analysis. This information has been added as well in the manuscript on page 8.

  1. According to the TEM results (Figure 2), the resulting catalyst particles have a fairly dense aggregation. From the point of view of catalysis efficiency, such results can adversely affect the properties of the resulting material. Authors should comment on this point and suggest a solution to this problem or explain its absence.

Response: Thank you for this professional and valuable question. The observed spherical-like morphology with small grain size agglomeration is in good line with XRD and BET results, which will positively affect the movement of absorbed oxygen and active sites advantageous for the oxidation reaction. This comment has been added in the manuscript on page 8.

  1. Isn't nitric oxide (II) released in large quantities a problem for the practical application of the created catalyst? I think the authors need to comment on this.

Response: Thank you for this professional and valuable question. Actually, It’s not nitric oxide (NO) but nitrogen dioxide (NO2). The reaction mechanism has been revised on page 19, as well as the Figure 11.

Round 2

Reviewer 1 Report

The authors have answered my comments and enclosed most of them in the revised version of the manuscript. However, some questions must be corrected before the paper could be accepted. The authors report the uncertainty of conversion and the apparent activation energy in page 9, but uncertainty of these parameters is vanished in the Abstract and Conclusions sections. However, uncertainty of selectivity to CO2 and NO2 in page 8 is not quoted. Finally, it seems to me that Figure 9b (Arrhenius plot) is not correct. It is not possible that the fitted k line was the one painted in this figure. The locus of the fitted k line should be between the four experimental points, instead over only two of them. The fit should be revised, the apparent activation energy estimated again, and the graph plotted correctly.

Author Response

Comments to the Author

The authors have answered my comments and enclosed most of them in the revised version of the manuscript. However, some questions must be corrected before the paper could be accepted.

Response: We thank the reviewer for his/her comments.

All the remarks and suggestions have been considered to improve the manuscript’s quality.

  1. The authors report the uncertainty of conversion and the apparent activation energy on page 9, but uncertainty of these parameters is vanished in the Abstract and Conclusions sections. However, uncertainty of selectivity to CO2and NO2on page 8 is not quoted.

Response: Thank you very much for these insightful comments and suggestions. The Abstract, Conclusions, and the uncertainty of selectivity to CO2 and NO2 on page 8 have been revised accordingly.

  1. Finally, it seems to me that Figure 9b (Arrhenius plot) is not correct. It is not possible that the fitted k line was the one painted in this figure. The locus of the fitted k line should be between the four experimental points, instead over only two of them. The fit should be revised, the apparent activation energy estimated again, and the graph plotted correctly

Response: Thank you very much for this valuable comment. Figure 9b has been revised accordingly. The fit has been revised. The fitted k line is now between the four experimental points. The apparent activation energy estimated to be 33.30 ± 3 kJ mol-1.

Updated Figure 9b:

Figure 9 Variation of the reaction rate of DMF (a), and Arrhenius plot for the catalyst on the DMF reaction evaluated from data in the conversion range within 15% (b).
